# Nutritive Value of *Ajuga iva* as a Pastoral Plant for Ruminants: Plant Phytochemicals and In Vitro Gas Production and Digestibility

Hajer Ammar [1,*,†], Ahmed Eid Kholif [2,†], Yosra Ahmed Soltan [3], Mohammad Isam Almadani [4], Walid Soufan [5], Amr Salah Morsy [6], Saloua Ouerghemmi [7], Mireille Chahine [8], Mario E. de Haro Marti [9], Sawsan Hassan [10], Houcine Selmi [11], Egon Henrique Horst [12,13] and Secundino Lopez [13]

1 Laboratoire de Systèmes de Production Agricole et Développement Durable "SPADD", UCAR, Ecole Supérieure d'Agriculture de Mograne, Mograne Zaghouan 1121, Tunisia
2 Dairy Science Department, National Research Centre, 33 Bohouth St. Dokki, Giza 12622, Egypt
3 Animal and Fish Production Department, Faculty of Agriculture, Alexandria University, Alexandria 21516, Egypt
4 Thünen Institute of Farm Economics, Bundesallee 63, 38116 Braunschweig, Germany
5 Plant Production Department, Faculty of Food and Agriculture Sciences, King Saud University, Riyadh 11451, Saudi Arabia
6 Livestock Research Department, Arid Lands Cultivation Research Institute, City of Scientific Research and Technological Applications, Alexandria 21934, Egypt
7 Laboratoire de Biotechnologie des Plantes, Banque Nationale des Gènes (BNG), ZI Charguia 1, Tunis 1080, Tunisia
8 Department of Animal, Veterinary and Food Sciences, University of Idaho, Moscow, ID 83844, USA
9 Gooding County Extension, College of Agricultural and Life Sciences, University of Idaho, 203 Lucy Lane, Gooding, ID 83330, USA
10 International Center for Agricultural Research in Dry Areas (ICARDA), Amman 11185, Jordan
11 Laboratoire des Ressources Sylvo-Pastorales, Institut Sylvo-Pastoral de Tabarka, Tabarka 8110, Tunisia
12 Department of Veterinary Medicine, Universidade Estadual do Oeste do Paraná, Campus de Cascavel, Guarapuava 85040-167, Paraná, Brazil
13 Departamento de Producción Animal, Universidad de León, 24007 León, Spain
* Correspondence: hjr.mmr@gmail.com or hajer.ammar@esamo.ucar.tn; Tel.: +216-96-721-222
† These authors contributed equally to this work.

**Abstract:** This study aims to evaluate the nutritive value of Ajuga iva (*A. iva*) harvested from three distinct altitude regions in Tunisia (Dougga, Mograne, and Nabeul). The chemical composition, phenolic concentration, gas production, and in vitro dry matter (DM) digestibility were determined. The highest concentrations of neutral detergent fiber (NDF) and acid detergent fiber (ADF) were for *A. iva* cultivated in Nabeul. In contrast, the highest crude protein (CP) concentration was observed in that cultivated in Mograne, and the lowest ($p < 0.01$) CP concentration was noted in that cultivated in Dougga. Additionally, the cultivation regions affected the concentrations of free-radical scavenging activity, total flavonoids, and total polyphenols ($p < 0.01$). The highest free-radical scavenging activity was observed with *A. iva* cultivated in Dougga and Mograne. The highest ($p < 0.05$) gas production rate and lag time were observed in *A. iva* cultivated in Mograne and Nabeul regions. DM digestibility differed between regions and methods of determination. The highest ($p < 0.01$) DM degradability, determined by the method of Tilley and Terry and the method of Van Soest et al., was for *A. iva* cultivated in Mograne and Dougga, while the lowest ($p < 0.01$) value was recorded for that cultivated in the Nabeul region. Likewise, metabolizable energy (ME) and protein digestibility values were higher for *A. iva* collected from Mograne region than that collected from the other sampling areas. In conclusion, the nutritive value of *A. iva* differed between regions. Therefore, care should be taken when developing recommendations for using *A. iva* in an entire region. Season- and region-specific feeding strategies for feeding *A. iva* are recommended.

**Keywords:** *Ajuga iva*; chemical composition; nutritive value; unconventional feeds; phenolic; growing conditions

## 1. Introduction

Inadequate feed supply is one of the significant challenges facing ruminant livestock producers, making exploring new feeds a premium issue for successful animal production [1]. Evaluating the nutritive value of unconventional feeds is essential before feeding them to animals. However, the nutritive value of plants depends on many factors and may differ for the same plant under different conditions. Reasons for that variability can be classified as intrinsic (variety, chemical composition) or extrinsic factors (growing conditions, storage, etc.) [2]. In addition, soil types, environmental conditions, geographical areas, and many more characteristics affect the nutritive value of feeds [3].

Plants contain secondary metabolites, including flavonoids, phytosterols, tannins, saponins, alkaloids, terpenoids, cyanogenic glycosides, etc., with multiple biological activities [4]. The concentration of secondary metabolites in plants depends on growth stage, soil type, etc. Soil type plays a vital role in determining the concentration and type of plant secondary metabolites. It is the matrix through which potential secondary metabolites are adsorbed and pass [5]. The activities of plant secondary metabolites in the soil are strongly linked with the soil's physical, chemical, biological, and physicochemical properties, which in turn affect adsorption and degradation [6]. *Ajuga iva* (L.) Schreber (Lamiaceae) (*A. iva*) is a plant that has been used in traditional medicine due to its anti-inflammatory, antifungal, antimicrobial, antifebrile, and anthelmintic activity [7]. *A. iva* contains polyphenolic compounds with antioxidant properties [8]. Its extract has been used traditionally as a diuretic, cardiac tonic, hypoglycemic, or a cure for fever. It exhibits a high stimulating effect on animal protein synthesis [9,10]. Chemical studies on *A. iva* have revealed the presence of several flavonoids, tannins, terpenes, and steroids [11]. The natural presence of bioactive compounds in the plant suggests the possibility of its use in animal feed to alter ruminal fermentation [12,13]. Recently, Bouyahya et al. [14] compared the volatile compounds of *A. iva* essential oils at three developmental periods, and noted that phenological stages significantly affected the volatile compounds resulting in different biological properties. They identified 28 volatile compounds in *A. iva* essential oils at the three developmental periods, with carvacrol, methyl chavicol, and octadecane among the significant compounds with different concentrations in each developmental period.

Additionally, the antioxidant, antibacterial, and antifungal properties of *Ajuga* were significantly affected by the concentrations of total phenolics and flavonoids [9]. Thus, the hypothesis of the present study depends on the possibility of a relationship between the nutritional value of *A. iva* feed materials and the sites of cultivation, as well as their content of phenolic substances, which may have an impact on their in vitro gas production and digestibility. Therefore, the present trial was undertaken to compare in vitro the nutritive value of *A. iva* at different sites (Dougga, Mograne, and Nabeul) in Tunisia.

## 2. Materials and Methods

### 2.1. Sampling Source of Ajuga iva (A. iva)

Approximately 1000 g of naturally cultivated mature *A. iva* parts (leaves and small stems) were randomly collected in Spring 2018 from three different sites in Tunisia: Nabeul (latitude 36°22′556″ N longitude 11°40′4581″ E), Dougga (latitude 36°25′94″ N; longitude 9°13′05″ E), and Mograne (latitude 36°40′920″ N and longitude 10°27′918″ E). These sites were selected to represent the majority of Mediterranean conditions in Tunisia. The texture of the soil in Nabeul was sandy, it was silty in Mograne, and vertisol (very fertile and rich in clay) in Dougga. The three regions are situated in semi-arid areas (precipitation ranges between 400–600 mm/year). The plant samples were air-dried at room temperature (40 ± 2 °C) for one week, ground by a Retsch blender mill (Normandie-Labo, 7210, type ZM1, Lintot, France), and sieved through a 0.5 mm mesh screen to obtain a uniform particle size. The ground substrates were bagged and stored at room temperature until the chemical analysis and in vitro experiments.

## 2.2. Chemical Analysis

All *A. iva* samples were analyzed in triplicate for dry matter (DM, method ID 934.01), ash (method ID 942.05), ether extract (EE, method ID 920.30), and crude protein (CP, method ID 984.13) content following the methods of AOAC [15]. Neutral detergent fiber (NDF), acid detergent fiber (ADF), and acid detergent lignin (ADL) were determined using an ANKOM2000 fiber analyzer [16] (ANKOM 2000, ANKOM Technology, Macedon, NY, USA) with the reagents described by Van Soest et al. [17]. Sodium sulfite, but not β-amylase, was added to the solution for NDF determination.

## 2.3. Phytochemical Analysis

For a detailed analysis of the bioactive components in *A. iva*, triplicate samples (1 g) were extracted with 20 mL of hydro-ethanolic solution (700 mL/L) according to the method described by Neffati et al. [18]. Extractions were carried out using maceration at room temperature for 24 h. The mixture was then filtered through Wattman No.1 filter paper (Bärenstein, Germany) and micro filter paper (Wattman, 0.45 μm). The resulting solutions were evaporated under vacuum at 40 °C using a rotavapor (Buchi Corporation R-210, New-Castle, DE, USA), and the yield (%) of extraction was determined. Samples were stored at 4 °C until use. The extract yield (%) was determined according to the below equation:

$$\text{Yield } (\%) = \frac{weight\ of\ dried\ extract\ (mg)}{weight\ of\ dried\ plant\ material\ (mg)} \times 100 \qquad (1)$$

The total phenolic (TP) content was determined by the Folin–Ciocalteu colorimetric method [19] with some modifications, using gallic acid as the standard. The method is based on reducing phosphotungstate–phosphomolybdate complex to blue reaction products [19]. The modified method is described briefly: *A. iva* leaf extract and the chosen standard (gallic acid) were dissolved at different concentrations, and 0.1 mL of each solution was mixed with 1 mL of Folin–Ciocalteu reagent (10%). The mixture was incubated for 5 min before adding 1 mL of 10% ($w/v$) $Na_2CO_3$. Prepared solutions were then diluted with 8.4 mL of deionized water and incubated in the dark at room temperature for 90 min. The absorbance of each sample and of the standard mixture was measured at 760 nm against the appropriate blank using a spectrophotometer (Jenway spectrophotometer monofaisceau UV/visible model 7315). The TP concentration was expressed as gallic acid equivalents per milligram of dry extract (mg GAE/mg DE).

Total flavonoid content (TF) was measured using a colorimetric method based on the formation of flavonoid [20]. Each diluted sample extract (0.25 mL) was added to 0.075 mL of $NaNO_2$ solution (7%) and mixed for 6 min before adding 0.15 mL of freshly prepared $AlCl_3$ solution ($6H_2O$, 10%). Catechin was used as a standard. After 5 min, 0.5 mL of 1 mol/L NaOH solution was added. The final volume was adjusted with distilled water to 2.5 mL, and thoroughly mixed; the absorbance of the mixture was determined at 510 nm. TF of dried *A. iva* leaf extract was estimated according to the calibration curve obtained by a series of concentrations of the catechin standard (0 to 700 μg/mL range). Samples were analyzed in triplicate, and results were expressed as catechin equivalents per mg dry extract (mg CE/mg DE).

## 2.4. Antioxidant Activity

The anti-radical activity (ARSA) of *A. iva* leaves was evaluated as the scavenging of the free anionic 2,2-diphenyl-1-picrylhydrazyl (DPPH) radical. At different concentrations, the sample solution (50 μL) or standard Trolox solution was added to 1 mL of 40 μM DPPH in methanol. The mixture was then shaken vigorously. After incubation (1 h), changes in color (from deep violet to dark yellow) were measured at 517 nm. The radical scavenging

activity for DPPH was evaluated by calculating the percentage of inhibition (PI) using the control and sample recorded absorbance (Abs):

$$\text{PI }(\%) = \frac{Abs\ control - Abs\ sampl}{Abs\ control} \times 100 \tag{2}$$

The half-maximal inhibitory concentration (IC50) was calculated from the linear equation of the curve obtained by projection of PI versus the respective sample concentrations.

*2.5. In Vitro Assays*

In vitro trials were carried out using two different methods: the in vitro dry matter digestibility, a gravimetrical method, and the in vitro gas production technique. In vitro dry matter digestibility analysis was performed in Spain (University of León, León, Spain) according to the technique proposed by Tilley and Terry [21] or by Van Soest et al. [22]. The in vitro gas production analysis was carried out in Tunisia at the Sylvo-Pastoral Institute of Tabarka.

2.5.1. In Vitro Dry Matter Digestibility (IVDMD)

Four mature Merino sheep with 49.4 ± 4.2 kg body weight (mean ± standard error) and fitted with a permanent ruminal cannula were used as inoculum donors to carry out in vitro incubations of the plant material. Animals were allowed 1 kg of Lucerne (Medicago sativa) hay (the traditional Mediterranean forage) once per day supported with ground maize and soybean meal (0.7 kg/100 kg live weight, 156 g CP/kg), and had free access to a mineral premix and fresh water. Sheep were cared for and handled by trained personnel in accordance with the Spanish guidelines for experimental animal protection (Spanish Royal Decree 53/2013 on the protection of animals used for experimentation or other scientific purposes). The experimental protocols were approved by the Institutional Ethics Committee on Animal Experimentation (ULE_014_2016) of Universidad de León and the Junta de Castilla y León (León, Spain). Ruminal liquid and solid parts were collected separately from each animal before morning feeding. The liquid part was collected by a stainless steel probe (2.5 mm screen) attached to a large-capacity syringe. The solid part was collected from the dorsal rumen sac through the cannula, and squeezed by hand. Liquid and solid parts were placed separately under anaerobic surroundings into pre-warmed thermo containers (39 °C) and were carried immediately to the laboratory. The two parts were blended at 1:1 (*v/v*) for 10 s, squeezed through four layers of cheesecloth, and retained in a water bath (39 °C) flushed under $CO_2$ until the inoculation took place.

The in vitro dry matter digestibility (IVDMD) was determined using the Ankom Daisy procedure [16], to which two different approaches were applied as proposed by Tilley and Terry, and Van Soest et al. [22]. Both techniques were carried out separately in different trials. A culture medium containing macro and micro mineral solutions, resazurin, and a bicarbonate buffer solution was prepared as described by Van Soest et al. [22]. The medium was kept at 39 °C and saturated with $CO_2$. Oxygen in the medium was reduced by adding a solution containing cysteine–HCl and $Na_2S$, as Van Soest et al. [22] described. Rumen fluid was then diluted into the medium at a proportion of 1:5 (*v/v*). *A. iva* samples (250 mg) were weighed into artificial fiber bags (size 5 cm × 5 cm, pore size 20 µm), sealed with heat, and placed in incubation jars. Each jar was a 5 L glass recipient with a plastic lid provided with a single-way valve to avoid the accumulation of fermentation gases. Each incubation jar was filled with 2 L of the buffered rumen fluid transferred anaerobically, closed with the lid, and the contents mixed thoroughly. The jars were then placed in a revolving incubator (Ankom Daisy Incubator, ANKOM Technology Corp, Macedon, NY, USA) at 39 °C, with continuous rotation to facilitate the effective immersion of the bags in the rumen fluid. After 48 h of incubation in buffered rumen fluid, samples were either subject to 48 h pepsin–HCl digestion as described by Tilley and Terry [21], or gently rinsed in cold water followed by extraction with a neutral detergent solution at 100 °C for 1 h as

described by Van Soest et al. [22]. According to Van Soest [23], the original method of Tilley and Terry is a measurement of the apparent in vitro digestibility (AIVD).

Treatment with the neutral detergent solution removed bacterial cell walls and other endogenous products and, therefore, the residuals can be considered a determination of the true in vitro digestibility (TIVD) of dry matter. The first stage of ruminal incubation (48 h) following the Goering and Van Soest technique corresponds to the determination of dry matter degradability (IVdeg). Each technique was performed in duplicate (two bags per sample) and repeated in three runs in different weeks, giving six observations per sample.

### 2.5.2. Kinetics of Gas Production

Rumen fluid was extracted from four mature slaughtered Queue Fine de l'Ouest sheep ($48.5 \pm 4.3$ kg body weight), collected in a thermos, and transported immediately to the laboratory where it was strained through various layers of cheesecloth and kept at 39 °C under a $CO_2$ atmosphere. A culture medium was prepared as described previously and the rumen fluid was diluted in the culture medium at the proportion 1:2 (*v:v*). Plant material samples (300 mg) were weighed in a glass syringe (capacity 100 mL), added to 30 mL of the culture medium, and incubated in a water bath. The volume of gas produced in the syringes was measured every 2 h (from 0 to 72 h). Data were fitted to the model proposed by France et al. [24]:

$$G = A\,[1 - e - c(t - L)] \tag{3}$$

where G (mL) denotes the cumulative gas production (GP) at time t; A (mL) the asymptotic gas production; c ($h - 1$) the fractional rate of gas production and L (h) is the lag time. Effective degradability (ED, g DM degraded/g DM ingested) for a given rate of passage ($k$, $h^{-1}$) was estimated following the approach derived by France et al. [24]. To calculate ED, a mean retention time of digesta in the rumen of 30 h was assumed, giving a rate of passage of 0.033 $h^{-1}$ (which can be found in sheep fed on a forage diet at maintenance level). The partitioning factor (PF) was calculated as the ratio between net GP and the degradation of the organic matter during 24 h, and was used as an indicator of microbial protein syntheses [25].

### 2.6. Calculations

Metabolizable energy (ME, MJ/kg DM) content was estimated using CP and EE contents (g/kg DM) and the volume of gas measured after 24 h of incubation (G24 in mL per 300 mg DM incubated), as described by Menke and Steingass [26]:

$$ME = 2.43 + 0.1206 \times G24 + 0.0069 \times CP + 0.0187 \times EE \tag{4}$$

The digested organic matter (DOM), protein values [dietary protein undegraded in the rumen (PDIA), true protein degraded in the small intestine (PDIN), and true protein absorbable in the small intestine (PDIE)], and net energy status (in terms of forage units for lactation (UFL) or meat production (UFV)) of Ajuga foliage were assessed according to the INRA [27] feed evaluation system. These were estimated from the feed characteristics (chemical composition and in vitro digestibility parameters) obtained in our study using the INRAtion software (V5, RUMIN'AL, Paris, France).

The partitioning factor (PF) was calculated as mg DM digested potential of degradability (D144)/mL gas production (A) [25], as an indicator of the efficiency of ruminal microbial protein synthesis.

### 2.7. Statistical Analysis

All data were analyzed by Tukey's test according to a split-plot design, with the whole plots arranged in a randomized block design. Statistics were carried out using the PROC GLM procedure of SAS (v. 9.2; SAS Institute Inc., Cary, NC, USA). The mean values of each parameter and the pooled standard error of the mean (S.E.M.) are reported in the tables. Differences between treatments were considered significant at $p < 0.05$ using Duncan's test.

## 3. Results

### 3.1. Chemical Composition and Phytochemical Contents

Table 1 shows the chemical composition of *A. iva* cultivated in different Tunisian regions. *A. iva* cultivated in Nabeul provided the highest ($p < 0.001$) values of Ash, NDF, ADF, and EE compared with the other regions, while that collected in Mograne had the highest ($p < 0.001$) CP value compared with the other two regions. On the other hand, the lowest ($p < 0.001$) values of CP and EE were observed for *A. iva* cultivated in the Dougga region.

**Table 1.** Chemical composition, active phytochemicals (g/kg dry matter), and anti-radical scavenging activity (μg/mL) of *A. iva* leaves cultivated in three different Tunisian regions.

| Items | Regions | | | S.E.M. | *p*-Value |
|---|---|---|---|---|---|
| | **Dougga** | **Mograne** | **Nabeul** | | |
| DM | 892 | 898 | 905 | 2.94 | 0.060 |
| Ash | 165 [b] | 155 [b] | 244 [a] | 3.07 | <0.001 |
| CP | 81.7 [c] | 134.5 [a] | 102.4 [b] | 2.27 | <0.001 |
| NDF | 279 [b] | 262 [b] | 332 [a] | 3.90 | <0.001 |
| ADF | 212 [b] | 202 [b] | 274 [a] | 4.34 | <0.001 |
| ADL | 50.1 | 51.6 | 46.2 | 2.03 | 0.229 |
| EE | 10.8 [c] | 11.2 [b] | 12.2 [a] | 0.072 | <0.001 |
| TF (mg CE [1]/mg DM [2]) | 0.34 [b] | 0.17 [c] | 0.93 [a] | 0.020 | <0.001 |
| TP (mg GAE [3]/mg DM) | 0.79 [a] | 0.61 [c] | 0.72 [b] | 0.013 | <0.001 |
| ARSA (μg/mL) | 485 [a] | 343 [b] | 71.2 [c] | 6.283 | <0.001 |

DM = Dry matter, CP = Crude protein, NDF = Neutral detergent fibre, ADF = Acid detergent fibre, ADL = Acid detergent lignin, EE = Ether extract, TF = Total flavonoid, TP = Total phenolic, ARSA = Anti-radical scavenging activity. [a,b,c] = letters within the same row, mean values not sharing a common superscript represent significant differences ($p < 0.05$), S.E.M. = Standard error of the mean. [1] CE = Catechin equivalents, [2] DM = Dry mater, [3] GAE = Gallic acid equivalents.

Results of TF, TP, and anti-radical scavenging activity "ARSA" of *A. iva* samples collected from different regions of Tunisia generally showed that Ajuga is a rich phytochemical plant (Table 1). Highly significant values ($p < 0.001$) of TP and ARSA were recorded in *A. iva* from Dougga, while those collected from Nabeul had the highest ($p < 0.001$) TF compared with *A. iva* from other regions. Leaves collected from Mograne had the lowest TF and TP, while samples collected from Nabeul resulted in the lowest ($p < 0.001$) ARSA compared with other plant samples.

### 3.2. In Vitro DM Digestibility (IVDMD)

As shown in Table 2, the highest values ($p < 0.05$) of in vitro DM digestibility (IVDMD) measured either by the Tilley and Terry or Van Soest et al. methods [23] were observed in *A. iva* collected from Mograne and Dougga, while leaves collected from Nabeul had the lowest ($p < 0.01$) nutrient digestibility values.

**Table 2.** In vitro dry matter degradability of *Ajuga. iva* leaves cultivated from three different regions in Tunisia.

| Parameters | Regions | | | S.E.M. | *p*-Value |
|---|---|---|---|---|---|
| | Dougga | Mograne | Nabeul | | |
| AIVD (g/kg) | 699 [a,b] | 741 [a] | 631 [b] | 1.58 | 0.008 |
| IVdeg (g/kg) | 577 [a] | 599 [a] | 491 [b] | 1.69 | 0.009 |
| TIVD (g/kg) | 776 [a] | 793 [a] | 679 [b] | 1.04 | <0.001 |
| ED | 451 [a] | 492 [a] | 392 [b] | 1.18 | 0.003 |

AIVD = Apparent in vitro dry matter digestibility, IVdeg = in vitro degradability of dry matter, TIVD = true in vitro digestibility, ED = effective degradability. [a,b] = letters within the same line, mean values not sharing a common superscript represent significant differences (*p* < 0.05), S.E.M. = Standard error of the mean.

### 3.3. Gas Production Kinetics and Energy Status

The gas emitted from leaves of *A. iva* collected from three different regions incubated at different times (0–72 h) is illustrated in Table 3. *A. iva* collected from Nabeul had the lowest (*p* < 0.05) GP at 8, 16, and 24 h incubation times, the lowest yield of GP calculated at 24 h (GY24), and lowest average rate (AR) of GP compared with those collected from Dougga and Mograne regions.

**Table 3.** In vitro cumulative gas production (mL/g DM) and gas kinetics through 72 h of incubation of *Ajuga. iva* cultivated in three different regions in Tunisia.

| Items | Regions | | | S.E.M. | *p*-Value |
|---|---|---|---|---|---|
| | Dougga | Mograne | Nabeul | | |
| 8 h | 112 [a] | 113 [a] | 100 [b] | 2.3 | 0.012 |
| 16 h | 163 [a] | 167 [a] | 147 [b] | 1.9 | <0.001 |
| 24 h | 226 [a] | 210 [b] | 199 [b] | 2.9 | 0.002 |
| 36 h | 286 | 250 | 256 | 10.9 | 0.119 |
| 48 h | 312 | 277 | 279 | 11.4 | 0.122 |
| 72 h | 323 | 292 | 291 | 12.3 | 0.189 |
| A | 344 | 295 | 311 | 17.3 | 0.209 |
| c | 0.046 | 0.054 | 0.046 | 0.003 | 0.202 |
| GY24 | 231 [a] | 215 [a,b] | 204 [b] | 4.1 | 0.012 |
| AR | 11.5 [a] | 11.6 [a] | 10.2 [b] | 0.18 | 0.002 |
| PF | 2.26 | 2.69 | 2.22 | 0.125 | 0.069 |

A = Asymptotic gas production (mL/g DM incubated), c = Fractional rate of fermentation (/h), GY24 = Volume of fermentation gas produced at 24 h incubation (mL/g DM incubated), AR = Average gas production rate (mL/g DM per h), PF = Partitioning factor. [a,b] = letters within the same row, mean values not sharing a common superscript represent significant differences (*p* < 0.05); S.E.M. = Standard error of the mean.

Table 4 shows the ME, DOM, net energetic values (UFL and UFV), and protein values (PDIA, PDIN, PDIE) of *A. iva* leaves cultivated from three different regions. *A. iva* cultivated in Nabeul showed the lowest (*p* < 0.001) ME, UFL, and UFV values compared with samples collected in Dougga and Mograne. However, the highest (*p* < 0.001) PDIA (41.7 g/kg DM), PDIN (84.7 g/kg DM), and PDIE (87.7 g/kg DM) were recorded in *A. iva* cultivated in Mograne, compared with those cultivated in Nabeul and Dougga. As compared with those cultivated in other regions, samples of *A. iva* cultivated in Mograne tended to have the highest PF values (*p* = 0.06).

**Table 4.** Metabolizable energy (MJ/kg DM), digested organic matter (g/kg DM), energetic value (kg DM), and protein value (g/kg DM) of *A. iva* leaves cultivated from three different regions in Tunisia.

| Parameters | Regions | | | S.E.M. | *p*-Value |
|---|---|---|---|---|---|
| | Dougga | Mograne | Nabeul | | |
| ME (MJ/kg DM) | 8.59 [a] | 8.51 [a] | 8.08 [b] | 0.064 | 0.003 |
| DOM (g/kg) | 673 | 670 | 664 | 0.44 | 0.449 |
| UFL (MJ/kg DM) | 0.71 [a] | 0.70 [a] | 0.61 [b] | 0.006 | <0.001 |
| UFV (MJ/kg DM) | 0.64 [a] | 0.63 [a] | 0.54 [b] | 0.008 | <0.001 |
| Protein values (g/kg DM) | | | | | |
| PDIA | 25.7 [c] | 41.7 [a] | 31.7 [b] | 0.81 | <0.001 |
| PDIN | 51.3 [c] | 84.7 [a] | 65.0 [b] | 1.44 | <0.001 |
| PDIE | 73.0 [b] | 87.7 [a] | 73.3 [b] | 0.64 | <0.001 |

ME = Metabolizable energy, DOM = Digested organic matter, UFL = Net energy for lactation, UFV = Net energy for meat production, PDIA = Dietary protein undegraded in the rumen, PDIN = True protein digested in small intestine, PDIE = True protein absorbable in the small intestine. [a,b,c] = letters within the same row, mean values not sharing a common superscript represent significant differences ($p < 0.05$), S.E.M. = Standard error of the mean.

## 4. Discussion

### 4.1. Chemical Composition

The CP concentration of *A. iva* ranged between 8.2 and 13.5%, which is within the acceptable range reported for different foliage plants [28], and was above the minimum threshold of 80 g/kg DM required for rumen microbial growth and activity [29]. *A. iva* cultivated in Dougga had the lowest CP concentration ($p < 0.05$), while that cultivated in Mograne had the highest CP (13.5%). Irrespective of the region, based on CP, it appears that leaves of *A. iva* were at least comparable in value to most traditional Mediterranean legume forages such as lucerne hay [30]. Therefore, the significant contribution of such pastoral plants would suggest their potential for overcoming feed limitations for ruminant livestock in Mediterranean regions, especially during the drought season, justifying their use to complement poor-quality pastures and crop residues [30]. However, it is supposed that some nitrogenous compounds are encrusted in the cell wall structure [31], and consequently, the utilization of CP by animals may not be as high as expected. Thus, the chemical composition of these browse species should not be the sole criterion for judging the relative importance of a particular species. Concerning cell wall fractions (NDF and ADF), *A. iva* cultivated in Mograne had the lowest values ($p < 0.05$), at 26% and 20% for NDF and ADF, respectively. The variability observed between cultivated regions could be due to differences in climatic conditions, soil types, soil fertility, agronomical management, and other environmental factors [30,32,33]. In this context, Mountousis et al. [34] reported that NDF and ADF content of forages were affected by the altitudinal zone and the season.

### 4.2. Bioactive Phytochemicals and Antioxidant Activity of A. iva

Our present study shows that secondary metabolites (types and concentrations) varied widely with the site of *A. iva* cultivation. The environmental conditions across the three collection sites are the most probable causes of variations in the plant phytochemicals [35]. In the present experiment, these differences also resulted in variations of antioxidant activity. The differences between regions are related to many factors including differences in meters above sea level, soil type, soil chemical composition, erosion status, management systems, and other related aspects [33,36]. Moreover, the differences in AA between regions and extraction methods could be associated with differences in active ingredients due to different concentrations of phenolic compounds in *A. iva* [14].

Free-radical scavenging is one known mechanism by which antioxidants inhibit lipid oxidation [9]. In the present study, the AA differed between regions and from other studies

examining the leaves of *A. iva* harvested in Tunisia [9]. In the current study, we used ethanolic extraction; thus, the solvent extraction method seems to be the main reason for the differences between studies [37]. Extracts with high polarities, such as ethanol, give better results than weakly polar solvents such as petroleum ether or methanol. The primary function of plant secondary metabolites is defense against different environmental threats. Therefore, concentrations of plant secondary metabolites are expected to differ between cultivation zones. Extraction is the foremost step for recovering and isolating phytochemicals from plant materials, and the concentration of phytochemicals in plants depends on plant samples' physical properties and the solvent's polarity [34,35]. Extraction efficiency is affected by the chemical nature of phytochemicals, the extraction method, and the solvent used [5,38]. The sensitivity of the chemical method used to quantify the phenolic compounds and the nature of the standard can affect concentrations in the same sample. Makni et al. [9] observed that the extraction yield of *A. iva* differed between methanol, aqueous, hexane, and chloroform extractions. For *A. iva*, Bendif et al. [39] observed different concentrations of total phenolics and free-radical scavenging activity with different extraction methods (acetone, ethanol, and water). Ouerghemmi et al. [40] compared the phenolic composition and antioxidant properties of methanol and ethyl acetate extracts from leaves of *Rosa canina*, *Rosa sempervirens*, and *Rosa moschata* collected from different Tunisian regions and observed differed yields. Higher phenolic compounds indicate higher antioxidant activity (i.e., low free-radical scavenging activity).

Phenolic compounds are critical components in plant samples, and their ability to scavenge free radicals is due to their hydroxyl groups [35]. The highest free-radical scavenging activity was observed for *A. iva* cultivated in Dougga and Mograne. It has been proven that levels of total phenols and flavonoids are high when the living environment of the plant is not appropriate. In this case, the plant promotes the synthesis of secondary metabolites to adapt and survive.

### 4.3. In Vitro DM Digestibility and Kinetics of Gas Production

DM degradability differed among plant samples cultivated in different regions. The in vitro DM digestibility measured by the Van Soest et al. method [22] (TIVD) or by Tilley and Terry's method [21] (AIVD) for *A. iva* from different regions was within the range (36 to 69%) of in vitro DM digestibility observed for most browse plants [41]. The digestibility of DM determined using Goering and Van Soest's method was high for *A. iva* cultivated in Mograne, while the lowest value was recorded for that cultivated in the Nabeul region. The different results obtained by different methods of DM digestibility determination could be related to the conditions of each determination method. In vitro methods such as in vitro digestibility and gas production measurements are more reliable for detecting inhibitory compounds in feeds, because these compounds are likely to affect the activity of rumen microbes in a closed system [42,43]. As previously observed by Ammar et al. [44], using the in vitro gas production technique is preferred to other in vitro methods for estimating digestibility [26]. Moreover, in vitro gas production is very suitable for assessing the biological activity of tannins and other anti-nutritional factors affecting the digestibility of browse plants [44,45]. In the present experiment, *A. iva* cultivated in Nabeul showed the lowest values of AIVD, TIVD, degradability potential, and effective degradability compared with samples cultivated in Dougga and Mograne.

The kinetics of the in vitro GP differed between regions. Gas production is a good indicator of the ruminal fermentability of feeds [46,47]. It depends mainly on the degradability of soluble components in the incubated substrates, and the partitioning of fermented substrates to volatile fatty acids and microbial biomass production [25,48]. During the first 24 h of incubation, *A. iva* from Dougga and Mograne regions produced higher gas levels, a higher average rate of gas production, and higher gas yields at 24 h; however, the asymptote and the rate of gas production were not significantly affected, indicating different fermentability between *A. iva* from different regions. Differences could be due to variations in the chemical composition and nutrient degradability [49] of the *Ajuga*

cultivated in different zones. In the present experiment, the asymptotic gas production followed the same trend as OM and CP content and in vitro digestibility, conversely to the fiber content in *A. iva*, which confirms the results obtained by Ammar et al. [44,50]. They observed significant positive correlations between in vitro digestibility, GP parameters, and CP content, and negative correlations with NDF, ADF, and lignin contents. Furthermore, other factors including non-soluble carbohydrate fractions and phytochemicals affect the production of gases [51]. In the present experiment, the insignificantly different partitioning factor indicates similar efficiency of ruminal microbial protein synthesis [25].

The higher NDF and ADF concentrations in *A. iva* collected from Nabeul may be the main reason for the low degradability revealed in the GP experiment. The observed greater ED of *A. iva* cultivated in Dougga and Mograne, compared to that in Nabeul, is an indicator of how well it can be utilized by ruminants. Differences in ED may be attributed to chemical composition, particularly the structural and non-structural protein and carbohydrate fractions [52–54].

The low values of UFL and UFV indicate low energy availability for milk and meat production for animals consuming *A. iva* cultivated in Nabeul, compared to those in the Dougga and Mograne regions [27]. Moreover, the measured parameters of protein value indicate that *A. iva* cultivated in Dougga and Nabeul had lower nutritive protein value compared to that cultivated in the Mograne region [27]. Greater concentrations of protein undegraded in the rumen but truly digestible in the small intestine, as well as true protein absorbable in the small intestine when rumen fermentable energy is limited, are good indicators of high nutritive value and are important from a nutritional view as lower degradability at the beginning of incubation indicates greater bypass protein that can be utilized in the duodenum [55,56]. Microorganisms could more easily attach to better degradable protein in the rumen and reflect greater protein solubility [57].

## 5. Conclusions

Based on the chemical composition and the in vitro digestibility results, it seems that *A. iva* could be successfully used to complement protein deficiencies in the diet of ruminants during periods of feed scarcity. The nutritive value of *A. iva* greatly varies between geographical zones, suggesting a need for season- and region-specific feeding strategies. Further studies are needed to evaluate its palatability and demonstrate its efficacy in vivo. Studies are ongoing examining other biochemical activities of *A. iva* to demonstrate its medicinal properties.

**Author Contributions:** Conceptualization, H.A.; methodology, H.A., A.E.K., H.S. and S.L.; software, H.A.; validation, H.A., A.E.K., Y.A.S., A.S.M. and M.I.A.; formal analysis, A.E.K., S.O. and S.L.; investigation, H.A., A.E.K., Y.A.S. and A.S.M.; resources, H.A.; data curation, H.A., A.E.K., Y.A.S., S.O., S.H., E.H.H. and S.L.; writing—original draft preparation, H.A. and A.E.K.; writing—review and editing, H.A., A.E.K., Y.A.S., A.S.M., S.H., M.C. and M.E.d.H.M.; visualization, H.A., A.E.K., M.C., M.E.d.H.M., W.S. and S.L.; supervision, H.A. and M.I.A.; project administration, H.A.; funding acquisition, W.S. All authors have read and agreed to the published version of the manuscript.

**Funding:** This research received no external funding.

**Institutional Review Board Statement:** The study was conducted in accordance with the Declaration of Helsinki, and approved by the Institutional Review Board (or Ethics Committee) of University of Leon, Spain. The protocol was designed and all practices were performed according to the Directives 2010/63/EU of the European Parliament and of the Council of 22 September 2010 on the protection of the animals used for scientific purposes.

**Informed Consent Statement:** Not applicable.

**Data Availability Statement:** The data presented in this study are available on request from the corresponding author.

**Acknowledgments:** The authors want to thank the Researchers Supporting Project number (RSP2021/390), King Saud University, Riyadh, Saudi Arabia.

**Conflicts of Interest:** The authors declare no conflict of interest.

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
