# Peer review of "Nutritive Value of Ajuga iva as a Pastoral Plant for Ruminants: Plant Phytochemicals and In Vitro Gas Production and Digestibility"

_agriculture, doi:10.3390/agriculture12081199_

Round 1
Reviewer 1 Report
Dear author:
Please consider the comments.
Sincerely,

Author Response
Dear Reviwer, many thanks for your comments on our manuscript. We have taken into consideration all of them. Also English was reviwed by two native english speaker.
Herein (attached) the new version modified as suggetsed by you and by the other reviwer. Responses to your question are below illustared (marqued in yellow)
Many thanks for your patience
Pr. Hajer Ammar
Comments to the Author
Dear author:
Please consider the comments.
Sincerely,
Specific comments/Repsonses:
Abstract
L34: “organic matter (OM)” not found in the results section. The authors reported in term of “Ash”, which the OM could be calculated from 100-Ash. Rewrite please!
Done
L36-37: delete “The lowest (P<0.01) concentrations of NDF and ADF have been observed in A. iva cultivated Mograne, while”
Done
L41-42: delete “The highest (P<0.01) asymptotic gas production, and digestible OM have been observed for A. iva cultivated in Dougga.” Because, the asymptotic gas production not different (P=0.209) among sites and the digestible OM not found in the results section.
Done
L44-47: Change “The highest (P=0.009) DM degradability determined using the Tilley and Terry method was for A. iva cultivated in Mograne, while what cultivated in Nabeul had the lowest values. DM digestibility determined using Goering and Van Soest method was high (P<0.01) for A. iva cultivated in Mograne while the lowest value was noted for that cultivated in the Nabeul region.” to
“The highest (P<0.01) DM degradability determined from the method of Tilley and Terry and the method of Goering and Van Soest was for A. iva cultivated in Mograne and Dougga, while the lowest (P<0.01) value was noted for that cultivated in the Nabeul region.”
Done
L48: Please rewrite the sentence.
Sentence is rewriten
Introduction
Clearly expressed.
Thanks
Materials and Methods
Please, provide the information on animal ethic statement or guidance.
Answer: The etic guidance is provided
L95: Please add “Year”
The year of the samplig is added "2018"
L95-96: Add the detail on latitude and longitude of each site.
Latitude and longetude for each sampling region are added
L98: Please, provide the information on soil type of each site.
Informations are added
L99-100: Please rewrite the sentence.
L102: Add the model of blender mill.
Done
L110: Add “(ANKOM 2000, ANKOM Technology, Macedon, NY, USA)”
Detail is added
L112: Add the (model, company, city, country) of spectrophotometer.
Response: Ths phrase was deleted because an error occured before and the even data concerning the use of this spectro is not presneted, thus it was an error and the sentence as added before, now we delet it
L120: Add the “company name and city” of the rotary evaporator.
The company name is Buchi and the city is USA,
Company name and all details are added
L130: Please change “ml” to “mL”, and check throughout the manuscript.
Done
L133: Add the model of spectrophotometer.
Model is added
L136: Please change “0.25 ml…” to “Each diluted sample extract (0.25 mL) was…”
Done
L158: Please use this sentence “Four mature Merino sheep were 49.4 ± 4.2 kg body weight (mean ± standard error), and fitted with…”
Done
L170: under CO2 flushed…
Done
L172: Change “(IDMD)” to “(IVDMD)”, and check throughout the manuscript.
Done
L186-187: Please check the model of Ankom Daisy Incubator, and rewrite to “(Daisy Incubator…., ANKOM Technology, Macedon, NY, USA)”
Reference is added
L201: Rumen fluid was extracted from four mature slaughtered sheep…
Done
L202: Change “(48.5±4.3 kg SE body weight)” to “(48.5±4.3 kg body weight)”
Done
L203: Change “immediately o the” to “immediately to the”
Done
L225: “INRA [20]” to “INRA [58]”
It was an error in the code, now all references are reviwed and rectified. Changement of codes is Done
Results
Table 1: Change “Items.” to “Items”, and move to the left margin.
Done
L250: What is mean?
Answer: ARSA: anti-radical scavenging activity, abbreviation and is synonym is added
L259, 260: Change “(IDMD)” to “(IVDMD)”
Done
L261, L263: Change “(P < 0.05)” to “(P < 0.01)”
Done
L271-274: Please use this sentence “A. iva collected from Nabeul had the lowest (P<0.05) GP at 8, 16, and 24 h of incubation times, the yield of GP calculated at 24h (GY24), and average rate (AR) of GP compared with those collected from Dougga and Mograne regions.
Done
L281: Change “showed” to “shows”
Done
Table 4: Please move “Parameters” to the left margin and move “Regions” to the center.
Done
L264, 275, 290: Change “Ajuga. iva” to “A. iva”
Done
For the parameter lists in all Tables, please use the abbreviation as shown in M&M section, and described in the footnote.
Done
Discussion
L303: Change “Medeetraian regions” to “Mediterranean regions”
Done
L315: Change “A. iva” to “A. iva”
Done
L380-385: Not clear rewrite please!
Revision is done and we decided to delete this paragraph was deleted because it does not bring some new thing as compared with the previous.
Conclusion
L433, 437: delete “leaves”
Conclusion is reviewed
References
The references are incomplete, please check carefully!
All references are reviewed and coded according to the text

Reviewer 2 Report
conclusion drawn from study need to be improved as suggested

Author Response
Dear Reviwer, many thanks for your comments on our manuscript. We have taken into consideration all of them and we rectified as suggested.
Your comments on our first conclusion: "Information on palatability of plant is not known and also not studied in the present experiment. How authors can recommend for feeding of animals. Modified the conclusion" was taken into consideration and we changed the conclusion appropriately to the study .
Herein the reviw version (modified according to your comments and to those suggetsed by other reviwer).
Many thanks for your patience and for time you interst you gave to reviw the manuscript.
Pr. Hajer Ammar

Round 2
Reviewer 1 Report
Dear author:
Please consider the comments.
L264: “EE = Ether extract”
L269: change “anti-radical scavenging activity (ARSA)” to “ARSA”
Please Add “DOM” in Table 4.
For References, please use the italic for all scientific names.
Sincerely,
Author Response
Dear Reviewer, t,hanks for your revision and apreciated comments. All of them have been taken into consideration in our new version submitted herein.
Responsess to all comments are below listed:
L264: “EE = Ether extract”
Done
L269: change “anti-radical scavenging activity (ARSA)” to “ARSA”
Done
Please Add “DOM” in Table 4.
Done
For References, please use the italic for all scientific names.
All scientific names are in italic form
